# The role of point-of-care ultrasound in the assessment of schistosomiasis-induced liver fibrosis: A systematic scoping review

Eloise S. Ockenden[1], Sandrena Ruth Frischer[1], Huike Cheng[1], J. Alison Noble[2], Goylette F. Chami [1]*

**1** Big Data Institute, Nuffield Department of Population Health, University of Oxford, Oxford, United Kingdom, **2** Institute of Biomedical Engineering, Department of Engineering Science, University of Oxford, Oxford, United Kingdom

* goylette.chami@ndph.ox.ac.uk

**Data Availability Statement:** All data are in the manuscript and/or supporting information files.

## Abstract

### Background

Abdominal ultrasound imaging is an important method for hepatic schistosomiasis diagnosis and staging. Several ultrasound staging systems have been proposed, each attempting to standardise schistosomal periportal fibrosis (PPF) diagnosis. This review aims to establish the role of ultrasound in the diagnosis and staging of schistosomal PPF, and to map the evolution of ultrasound staging systems over time, focusing on internal validation and external reproducibility.

### Methods

A systematic search was undertaken on 21st December 2022 considering the following databases: PubMed/MEDLINE (1946-present), Embase (1974-present), Global Health (1973-present), Global Index Medicus (1901-present), and Web of Science Core Collection–Science Citation Index Expanded (1900-present) and the Cochrane Central Register of Controlled Trials (1996-present). Case reports, systematic reviews and meta-analyses, and studies exclusively using transient or shear-wave elastography were excluded. Variables extracted included study design, study population, schistosomal PPF characteristics, and diagnostic methods. The PRISMA-ScR (2018) guidelines were followed to inform the structure of the scoping analysis.

### Results

The initial search yielded 573 unique articles, of which 168 were removed after screening titles and abstracts, 43 were not retrieved due to full texts not being available online or through inter-library loans, and 170 were excluded during full text review. There were 192 remaining studies eligible for extraction. Of the extracted studies, 61.8% (76/123) of studies that reported study year were conducted after the year 2000. Over half of all extracted studies (59.4%; 114/192) were conducted in Brazil (26.0%; 50/192), China (18.8%; 36/192) or Egypt (14.6%; 28/192). For the species of schistosome considered, 77.6% (149/192) of studies

**Funding:** ESO received funding from the UKRI EPSRC as a DPhil studentship (2593890) associated with project (EP/S02428X/1). GFC received funding from the Wellcome Trust Institutional Strategic Support Fund (204826/Z/16/Z) and John Fell Fund as part of the SchistoTrack Project, Robertson Foundation Fellowship, and UKRI EPSRC Award (EP/X021793/1). Salary contributions to GFC were received from the Robertson Foundation Fellowship and the UKRI EPSRC Award (EP/X021793/1). The funders had no role in study design, data collection and analysis, decision to publish, or preparation of the manuscript.

**Competing interests:** The authors have declared that no competing interests exist

considered *S. mansoni* and 21.4% (41/192) of studies considered *S. japonicum*. The ultrasound staging systems used took on three forms: measurement-based, feature-based and image pattern-based. The Niamey protocol, a measurement and image pattern-based system, was the most used among the staging systems (32.8%; 63/192), despite being the most recently proposed in 1996. The second most used was the Cairo protocol (20.8%; 40/192). Of the studies using the Niamey protocol, 77.8% (49/63) only used the image patterns element. Where ultrasound technology was specified, studies after 2000 were more likely to use convex transducers (43.4%; 33/76) than studies conducted before 2000 (32.7%; 16/49). Reporting on ultrasound-based hepatic diagnoses and their association with clinical severity was poor. Just over half of studies (56.2%; 108/192) reported the personnel acquiring the ultrasound images. A small number (9.4%; 18/192) of studies detailed their methods of image quality assurance, and 13.0% (25/192) referenced, discussed or quantified the inter- or intra-observer variation of the staging system that was used.

## Conclusions

The exclusive use of the image patterns in many studies despite lack of specific acquisition guidance, the increasing number of studies over time that conduct ultrasound staging of schistosomal PPF, and the advances in ultrasound technology used since 2000 all indicate a need to consider an update to the Niamey protocol. The protocol update should simplify and prioritise what is to be assessed, advise on who is to conduct the ultrasound examination, and procedures for improved standardisation and external reproducibility.

### Author summary

Schistosomal periportal fibrosis is a form of liver fibrosis caused by a parasitic blood fluke, which is predominantly prevalent in sub-Saharan Africa, but also affects areas of South America and South-East Asia. Ultrasound imaging is the most used method to diagnose schistosomal periportal fibrosis because it is easy to transport, safe to use, and the fibrosis is highly echogenic. Many different systems to stage this fibrosis using ultrasound have been proposed, focused on measuring blood vessels in the liver or identifying features or patterns of echogenicity in the scan that might indicate a certain stage of disease. The Niamey protocol is the most recently proposed of these staging systems, having been published by the World Health Organisation in 1996, which contains guidance for staging of *S. mansoni* morbidity. This guidance was updated in 2000. In the same year, updated guidance for the staging of *S. japonicum* morbidity was published by the China Centre for Disease Control (CDC). Both *S. mansoni* and *S. japonicum*-focused protocols evolved from the Cairo protocol. Our study aimed to understand how ultrasound has been used to stage schistosomal periportal fibrosis and to investigate how different staging systems have been applied and validated. The studies found were first conducted in 1979, and were mostly conducted in Brazil, China and Egypt. The Niamey protocol was the most reported of the staging systems. There was widespread variation in the use, and therefore interpretation, of the Niamey protocol. Future research is needed to reduce the complexity of the Niamey protocol and to standardise how the Niamey protocol is used. More detailed ultrasound procedures, focused assessments on the ultrasound pattern element, and clinically-validated guidance on the severity of the ultrasound pattern elements is needed.

## Introduction

Schistosomiasis is a neglected tropical disease caused by a parasitic blood fluke. Three species of this blood fluke, *Schistosoma mansoni*, *S. japonicum* and *S. mekongi*, cause hepatic schistosomiasis, a severe morbidity of which is schistosomal periportal fibrosis (PPF) [1]. Schistosomal PPF comes about by mature female flukes laying eggs in the mesenteric venules that are swept back into the hepatic portal system. When the cells in the eggs differentiate, the eggs release an antigen that depending on host inflammatory and granulomatous responses can lead to fibrosis in the main portal vein and segmental branches [2]. Schistosomal PPF, depending on the degree and diffusion of fibrosis, ranges in clinical severity and can be categorised into stages. Accurate disease staging is needed to guide the monitoring of disease progression and stage-appropriate treatment interventions. Currently, abdominal ultrasound imaging is the most used non-invasive method for diagnosis and staging of PPF caused by schistosomiasis [3]. Point-of-care ultrasound imaging is inexpensive, portable, and carries no additional health risks, unlike methods such as computed tomography (CT) which expose the patient to harmful radiation. However, widespread variation in how ultrasound imaging is used for staging schistosomal PPF persists.

Standardisation of staging of schistosomal PPF has been attempted since the first published evaluation of ultrasound as a diagnostic method for the disease in 1988 [4], with several staging systems having been proposed in the last 45 years [5–9]. Each ultrasound staging system comprises different criteria for the diagnosis and staging of schistosomal PPF that broadly sit in three categories: measurement-, feature- and image pattern-based staging systems. Measurement-based staging systems rely on measurements of, for example, vessel diameters or organ size. Feature-based staging systems consider distinct characteristics such as liver surface texture or diffuse nodules visible in ultrasound images. Image pattern-based staging systems are similar to feature-based staging systems but less granular, using a holistic picture or overall liver characterisation. Liver disease tends to be split distinctly into focal and diffuse disease [10]. The three types of staging systems span to different extents the staging of both focal and diffuse schistosomal PPF. The two staging systems that have been proposed by the World Health Organisation (WHO) are the Cairo protocol in 1992 [7] and the Niamey protocol in 1996 [9]. The Niamey protocol is the current WHO-recommended staging system, despite only being updated once in 2000 since its proposition in 1996. Given the diversity of aims for its guidance and different existing schistosomal PPF staging systems, the Niamey protocol is a compromise between a measurement-based staging system and a qualitative image pattern-based staging system [11]. The intended use of the Niamey protocol was that diverse systems could be used to construct a score to stage schistosomal PPF. However, there is no recognised consensus on how a standardised ultrasound staging system for schistosomal PPF should be conceptualised and applied.

Two systematic reviews have been published between 2010 and 2022 concerning diagnostic methods for hepatosplenic schistosomiasis [11,12]. Tamarozzi et al. [12] looked at both diagnostic and treatment methods, and concludes that there is a lack of data on what factors should inform clinical management approaches for hepatosplenic schistosomiasis. El Scheich et al. [11] looked specifically at the Niamey protocol and its role in research since its proposition, advising that the measurements of portal branches should be eliminated from the protocol. However, neither of the published reviews aimed to track the development of different ultrasound staging systems and compare their usage and validation across time with no restriction. Both published reviews had a publication date restriction as part of their inclusion criteria, only considering studies published from 1990 and from 2001 respectively. In particular, the

review by el Scheich et al. has similar aims to the current review, but considers a time frame of 12 years, with the search having been carried out in 2012 and, therefore, requiring an update. Therefore, this scoping review evaluated how ultrasound technology usage for schistosomal PPF has changed over time, what methods were used for validation and reproducibility, and approaches to the future development of ultrasound-based staging systems for detecting schistosomal PPF.

## Methods

### Database search

The full review protocol was published on 6th February 2023 on the Open Science Framework website [13]. A systematic search was performed on 21st December 2022. Six databases were searched encompassing articles published from 1900 to present: PubMed/MEDLINE (1946-present), Embase (1974-present), Global Health (1973-present), Global Index Medicus (1901-present), and Web of Science Core Collection–Science Citation Index Expanded (1900-present) and the Cochrane Central Register of Controlled Trials (1996-present). The search terms used were as follows, given in the PubMed/MEDLINE format: (Schistosom* OR Bilharzia* OR snail* fever) AND (periportal* OR portal OR liver* OR Symmer's) AND (fibrosis) AND (ultraso* OR sono*). The search string was adapted to fit the styles of each database (S1 Table). Results were aggregated and duplicates partially removed using Covidence and manually reviewed [14]. The PRISMA-ScR (2018) guidelines informed the structure of the review (S1 Text) [15].

### Inclusion and exclusion criteria

Once the searches had been carried out, two reviewers (EO and SRF) screened the titles and abstracts of all search results. Following this, the full text of each study was obtained where available and all English language studies were screened by one reviewer (EO), and a random 10% were screened by a second reviewer (SRF). Studies in Mandarin were screened in full by one reviewer (HC). At the full-text stage, only studies in English or Mandarin, or with a published translation were considered.

Studies carried out in any year up to the search date were included. The studies must have taken place in a hepatic schistosomiasis endemic area, that is, areas endemic with *S. mansoni*, *S. japonicum* or *S. mekongi*. There were no restrictions on the study participants. Included studies must have involved staging of schistosomal PPF and ultrasound imaging had to be the, or one of the, diagnostic tools used. The data collected may have been qualitative or quantitative, or both. Animal studies were excluded, as well as case reports and systematic reviews and meta-analyses. All other study designs were eligible such as cohorts, cross-sectional designs, before-after studies, case-control studies, and randomised-controlled trials (RCTs). The study aim must not have been exploratory in nature with regard to the staging system used; the criteria for schistosomal PPF staging must have been defined before data collection. A publication from the same study was defined as a publication on the same population studied by the same study authors in the same year. In those cases, the publication that was included was the one with the largest sample size, or if all publications reported the same sample size then the study published first was included.

### Data extraction and analysis

A data dictionary was developed by listing variables relating to key study characteristics and those more tailored to the review aims. This was piloted and adapted when extracting the

studies if, for example, new variables were needed to capture the relevant information for the study aims. Microsoft Excel was used to record the extracted variables [16]. Extraction of the English language studies was performed in full by one reviewer (EO), and a random 10% of the extractions were verified by a second reviewer (SRF). Extraction of the Mandarin studies was performed in full by one reviewer (HC). The extracted data was exported to a comma-separated value (CSV) file from which summary statistics and plots were obtained using Python 3.9 [17].

Broadly, the extracted variables fell into four categories: key study characteristics, study population characteristics, variables relating to schistosomal PPF, and variables relating to the ultrasound imaging use in the studies. The key study characteristics were author/s, publication and study years, study duration, number of sittings/visits, country, locality, setting, aims, design, sampling strategy, species of schistosome, any coinfections considered, the study comparator/s if two or more diagnostic tests were compared, and if the study considered ultrasound imaging as the best available reference standard for staging of schistosomal PPF. With respect to the study population, the sample size was recorded, in addition to and the age range and sex of the participants.

Common aims of papers in this review were grouped as follows. Diagnostic agreement studies compared one method of measuring/grading fibrosis to another, which could be two ultrasound staging systems or an ultrasound staging system with another diagnostic test. A prevalence study was one exclusively concerned with measuring and reporting the prevalence of schistosomal PPF or schistosomiasis infection in a population. A study looking at clinical severity attempted to link grades of fibrosis to some clinical outcome, for example oesophageal varices. Risk factor studies aimed to link some factor(s), for example infection intensity, with severity of schistosomal PPF. An intervention study measured the effect of some intervention on schistosome infection or fibrosis grade. Studies looking at links between fibrosis and immune response were categorised as immunological profiling studies. Liver function studies looked at the effect of fibrosis on serum biomarkers indicating degree of liver function. A coinfection study specifically looked at the interaction between some infection and schistosomal PPF, whether in terms of ultrasound findings or clinical outcomes. A genetic or genomic study linked a specific single nucleotide polymorphism (SNP) or other aspects of sequencing to schistosomal PPF. Studies looking at pathogenesis were concerned with identifying developmental mechanisms of schistosomal PPF from infection to severe disease.

There are many ways to define when liver fibrosis is PPF or to determine whether it is caused by schistosomiasis. These definitions tend to arise from the different ultrasound staging systems that exist. Therefore, the definition of PPF was recorded, as stated by the study authors. The number of participants with PPF according to this definition was also recorded. If there were multiple study sittings, the baseline prevalence was recorded. Intervention details were recorded if stated including if any treatment, such as praziquantel, was given as part of the study.

The majority of the variables collected were with respect to the diagnostic criteria. The names of the staging systems cited and used in the study were recorded. If a staging system was not named, then grade descriptions used by the study were recorded. Stage and grade of fibrosis here were often used interchangeably; however, a grade often related to a specific staging system. The predefined options for the staging systems were the Homeida staging system [4], the Managil staging system [5], the Strickland classification [6], the Cairo protocol [7], the China Centre for Disease Control (CDC) guidance [8], and the Niamey protocol [9]. A translation into English of the staging system from the China CDC guidance is available in the supplements (S3 Text). If an alternative staging system without a published name or citation was used, and the grade definitions were clearly defined, then this was listed as unnamed. If it was unclear which staging system was used due to inconsistent citations or lack of grade

definitions, then this was listed as unclear. If measurements were used to stage fibrosis, the reference population and type of measurement thresholds applied were recorded. When a published staging system was used, it was recorded whether there were any adaptations made to the published criteria such as consolidating multiple grades into one category.

Looking more deeply into abdominal ultrasound data collection procedures, if stated, the views of the abdomen collected by ultrasound imaging were recorded. Technical specifications of the ultrasound technology used were recorded to gauge the technological advances and uptake of developing technology in this research context. The probe type–categorised as linear, convex or phased array [18]–and its frequency range were recorded if stated. The ultrasound machine brand and model was recorded as reported by the study authors. Details of the personnel performing the scans and interpreting the findings were noted, such as their occupation, training and experience, and whether they were blind to any clinical findings. Personnel details for acquisition and interpretation were collected separately to allow for the fact that these are two distinct tasks requiring different skill sets.

Variables relating to the internal validation and external reproducibility of the staging systems and ultrasound more broadly as a diagnostic method were collected. Here internal validation was defined as the use of a quality control technique after data collection to confirm the accuracy of the schistosomal PPF grade or an explicit assessment of the inter- or intra-observer variation using the method. Internal validation indicates whether the data can be reproduced within the exact same study design, context, and conditions. Whether any internal validation was performed and the how this was applied was recorded. The methods of measuring internal validation included re-scanning of participants or re-reading of images or videos either by the same observer or among different observers. The percentage of participant outcomes that were internally validated was recorded along with any other details relating to internal validation as provided by study authors. External reproducibility was defined as a study or comment on the generalisability of the staging system or diagnostic method used in contexts outside of their study. External reproducibility relates to the standardisation of the protocols across contexts and their testing in different study settings, which conveys the ability to repeat the study design at a different time or place and not introduce unexpected subjectivity in the staging system or diagnostic procedures.

## Results

### Screening process and study characteristics

Fig 1 shows a flowchart of the systematic review process with the number of studies excluded at each stage. The initial search yielded 1194 hits from which 192 studies were extracted. The number of hits from each database are detailed in S1 Table. The most common reasons for exclusion at the full-text stage were lack of schistosomal PPF grading, multiple papers being published on the same study, and only the abstract being available due to the abstract being published at a conference or included in a journal supplement.

A complete summary of the extracted variables is provided (S1 Dataset, citations in S2 Text). Fig 2 shows the geographical distribution of extracted studies, with the most common study countries being Brazil (26.0%; 50/192), China (18.8%; 36/192) and Egypt (14.6%; 28/192). Fig 3 shows the distribution in study year of the extracted studies, ranging from 1979–2019. There were only four studies carried out before the introduction of the first staging system, and there was an increase in the number studies after the proposition of the Cairo protocol. After 2000, studies were carried out consistently with 2–7 studies being carried out each year.

A shortened summary of key study characteristics can be found in Table 1. The majority of the studies concerned *S. mansoni* (77.6%); however, a large number of studies on *S. japonicum*

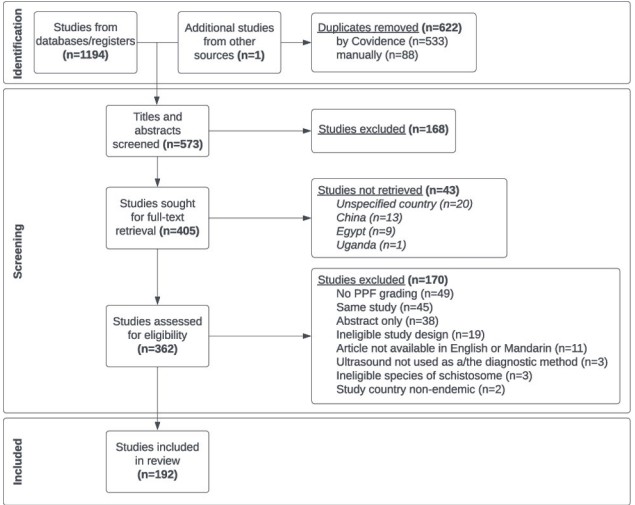

**Fig 1. PRISMA flowchart.** The screening and review process is split into three stages: 1) the identification of articles through an initial search, 2) the screening of articles following predefined inclusion and exclusion criteria and 3) the final included articles for data extraction.

were included (21.4%; 41/192). The most common aim of studies in this review was to assess diagnostic agreement, followed by measurement of prevalence and clinical severity. In the case of clinical severity studies, the outcomes considered were grade or bleeding of oesophageal varices and portal hypertension. Intervention studies were the most common study aim in studies concerning *S. japonicum* (29.3%; 12/41). The total number of study participants across all studies was 122,798. The age range of participants was 0–97; however, ultrasound may have been performed on a subset of participants, in particular it was unclear whether ultrasound was performed on children under five. The sex of participants was only restricted in four cases among all included studies (2.1%; 4/192). The three male-only studies selected participants based on

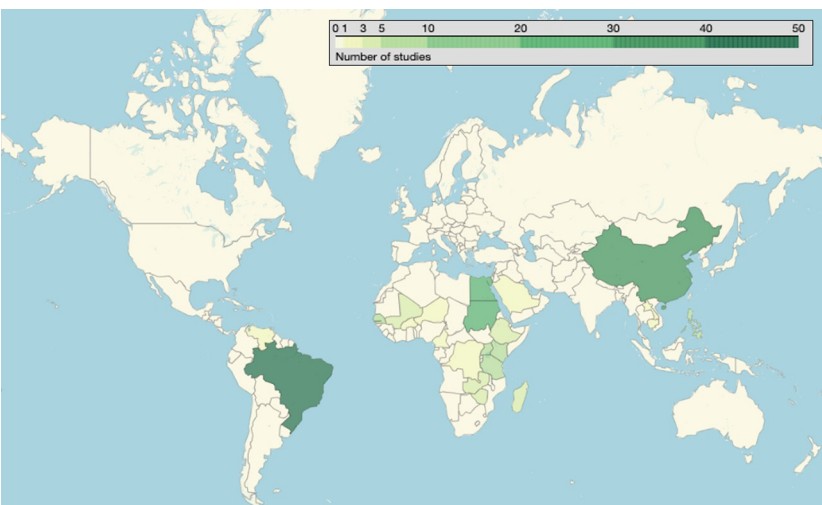

**Fig 2. Global distribution of studies.** A darker green colour indicates a higher number of studies conducted in that country. 190/192 studies reported study country. Open Street Map, which is licensed under an Open Database License ODbL 1.0, was used. The base layer was extracted here: https://www.openstreetmap.org/export. The terms and conditions of the copyright are provided here: https://www.openstreetmap.org/copyright.

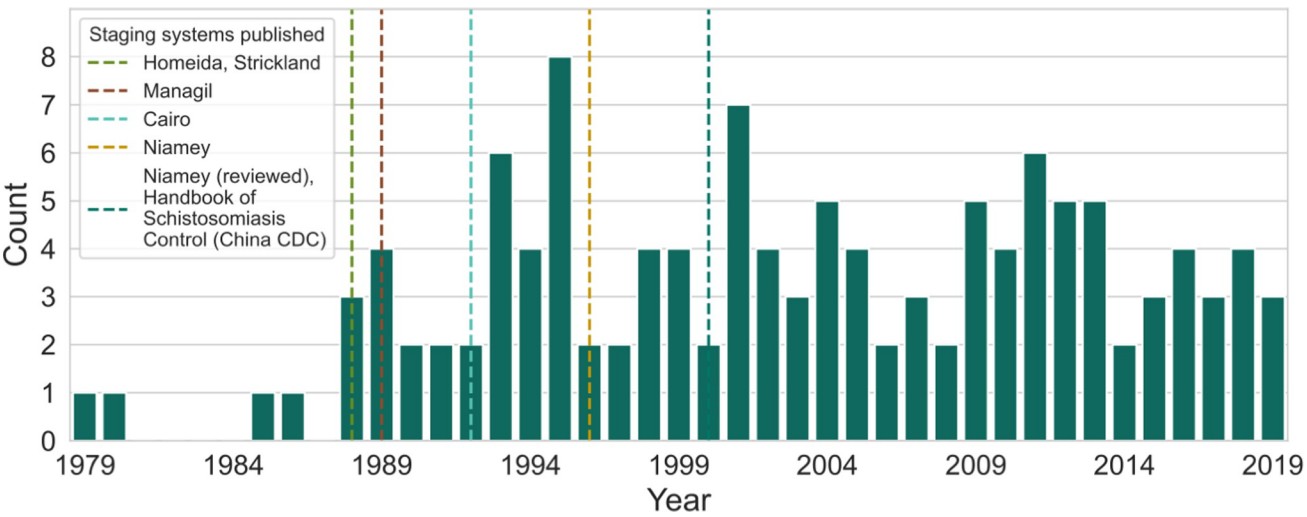

**Fig 3. Studies conducted by year (1979–2019).** Only studies that reported the study year are represented in this graph (64.1%; 123/192). The dashed line at the year 2000 represents both the Niamey revision and the China CDC guidance.

their occupation and the one female-only study was on the effect of different forms of contraceptives on liver function. The majority of hospital studies only included adults (81.8%; 54/66). Five study designs were represented, which were cross-sectional, before-and-after, cohort and case-control designs, and RCTs. All three of the case-control studies and both of the RCTs were conducted on *S. japonicum* in China. Some studies encompassed multiple designs such as the effect of an intervention being measured as a nested before-and-after study within a larger cohort. The mean percentage (std. dev.) of study participants that had schistosomal PPF according to ultrasound imaging was 52.3% (35.2). An intervention was offered to study participants in 45.3% of studies (87/192), and this involved praziquantel in 73.6% of these studies (64/87). The effect of a coinfection was considered in 17.7% of studies (34/192). The most common coinfections considered were *S. haematobium* (32.4%; 11/34), hepatitis C (32.4%; 11/34) and hepatitis B (29.4%; 10/34). These were not mutually exclusive.

## The role of ultrasound and evolution of staging systems

Different studies used different reference standards for the staging or diagnosis of schistosomal PPF. The vast majority used ultrasound as their reference standard (92.7%; 178/192). Some studies directly compared another diagnostic method to ultrasound grading (13.5%; 26/192), and when this happened the most common alternative was liver biopsy (46.2%; 12/26). Other diagnostic tests compared to ultrasound include transient or shear-wave elastography (26.9%; 7/26), and magnetic resonance imaging (MRI) (15.4%; 4/26). Only three of the 12 studies comparing to liver biopsy reported their year of study, however seven of the 12 were certainly carried out before 2000 due to being published before 2000. The most recent study found in the search that reported study year and performed biopsy reported a study year of 2011. For liver biopsy and ultrasound, overlap was found between positive schistosomal PPF diagnoses from the Strickland and Homeida staging systems and from biopsy in four studies. One of these studies showed a statistically significant association. By contrast, in three studies, no significant correlation was found between transient elastography (FibroScan) and the Niamey protocol image patterns. One study found that while transient elastography distinguished absent fibrosis from moderate or intense fibrosis, it was unable to distinguish moderate and intense fibrosis. This was done by grouping the Niamey protocol patterns into absent, moderate and

**Table 1. Key study characteristics.** Percentages are listed of the studies that specified this characteristic. The total number of studies extracted was 192. In all variables apart from *Schistosoma* species and sex of participants, categorisations are not mutually exclusive and therefore percentages may not sum to 100%.

| Study characteristic | Number reported | Category/Statistic | Value | Percentage |
|---|---|---|---|---|
| Study aim | 192 | Diagnostic agreement | 44 | 22.9% |
| | | Prevalence | 37 | 19.3% |
| | | Clinical severity | 29 | 15.1% |
| | | Intervention | 28 | 14.6% |
| | | Risk factors | 27 | 14.1% |
| | | Immunological profiling | 21 | 10.9% |
| | | Liver function | 17 | 8.9% |
| | | Co-infections | 17 | 8.9% |
| | | Genetic/genomic | 9 | 4.7% |
| | | Pathogenesis | 8 | 4.2% |
| Age of participants | 181 | Pre-school age children | 27 | 14.9% |
| | | School-age children | 94 | 51.9% |
| | | Adults | 165 | 91.1% |
| | | Range (years) | 0–97 | - |
| Sex of participants | 189 | No restriction | 185 | 97.9% |
| | | Male-only | 3 | 1.6% |
| | | Female-only | 1 | 0.5% |
| Sample size | 192 | Total | 122798 | - |
| | | Mean (std. dev.) | 640 (2168) | - |
| | | Range | 8–22488 | - |
| | | Interquartile range | 79.5–487.5 | - |
| Study duration | 107 | Mean (std. dev.) (months) | 24.9 (43.4) | - |
| | | Range (months) | 1–300 | - |
| Study sittings | 192 | One | 159 | 82.8% |
| | | Two | 16 | 8.3% |
| | | Three | 10 | 5.2% |
| | | More than three | 7 | 3.6% |
| Study setting | 180 | Community | 110 | 61.1% |
| | | Hospital | 66 | 36.7% |
| | | School | 10 | 5.6% |
| Study locality | 181 | Rural | 123 | 68.0% |
| | | Peri-urban | 9 | 5.0% |
| | | Urban | 55 | 30.4% |
| Study design | 192 | Cross-sectional | 165 | 85.9% |
| | | Before-and-after | 19 | 9.9% |
| | | Cohort | 11 | 5.7% |
| | | Case-control | 3 | 1.6% |
| | | Randomised controlled trial | 2 | 1.0% |
| Sampling strategy | 151 | Purposive | 78 | 51.7% |
| | | Random | 30 | 19.9% |
| | | Convenience | 19 | 12.6% |
| | | Census | 15 | 9.9% |
| | | Systematic | 8 | 5.3% |
| | | Stratified | 3 | 2.0% |
| *Schistosoma* species | 192 | *S. mansoni* | 149 | 77.6% |
| | | *S. japonicum* | 41 | 21.3% |
| | | *S. mekongi* | 2 | 1.0% |

intense fibrosis and comparing transient elastography results between pairs of these groups using a Kruskal-Wallis test.

Fig 4A shows a comparison between studies conducted pre-2000 and post-2000 with respect to the ultrasound staging system. The Niamey protocol was the most common staging system used (32.8%; 63/192) despite being the most recently introduced in 1996. Of the 63 studies using the Niamey protocol, 77.8% (49/63) only used the patterns element of the protocol to grade schistosomal PPF. No studies that concerned *S. japonicum* used the Niamey protocol due to the lack of guidance for this species in the protocol; the most popular protocols for this species were the Cairo protocol and the China CDC guidance (Fig 4B). The two studies concerning *S. mekongi* used the Niamey protocol. In 12.0% (23/192) of studies it was explicitly stated that the protocol that was cited/mentioned in the methods was modified from its original publication. In the majority of cases (69.6%; 16/23), this modification was a change to the definitions of the gradings such as consolidating multiple grades into one category. Five of 23 studies with modifications used the Niamey protocol image patterns.

## Method of ultrasound acquisition

The majority (78.6%; 151/192) of studies specified some aspect of the ultrasound technology that was used. Breakdowns of the ultrasound probe types used by time interval and by species of schistosome (*S. mansoni* and *S. japonicum*) are shown in Fig 5A and 5B respectively. Fig 6 shows the distribution of different manufacturers of ultrasound machine used in the studies. The most common manufacturers were Aloka (29.5%; 44/149), Siemens (20.1%; 30/149), and Hitachi (16.8%; 25/149), and these were not concentrated in particular countries. For the frequencies of the probe used, there was variation in the reporting of the studies. Some reported a single value of the frequency, some reported several discrete values and others reported a range. The most common frequency reported was 3.5MHz (50.0%; 96/192 studies). In terms of the views of the abdomen collected in the studies, 16.1% (31/192) of studies shared this information. The views provided among the studies that reported them included longitudinal and transverse subcostal views of left lobe, an oblique section of the main portal vein, and oblique intercostal views of right liver lobe. For studies that reported abdominal views, 19.4% (6/31) were acquiring image patterns using the Niamey protocol.

Just over half of studies reported any details of the personnel collecting and interpreting the ultrasound findings. Details on the personnel performing ultrasound examinations were reported in 56.2% (108/192) of studies. Of these studies, 52.8% (57/108) used only one person to acquire and interpret ultrasound images. The interpreter of ultrasound data was blind to other clinical findings, such as schistosome infection, in 46.3% (50/108) of these studies. Training of personnel was infrequently reported; 8.9% (17/192) of studies explicitly specified that ultrasound interpreters went through training as part of the study prior to the data collection. Under half (38.0%; 73/192) of studies specified the occupation of the ultrasound practitioner/ interpreter, of which the most common occupations were physicians of unspecified specialism (32.9%; 24/73) or sonographers (32.9%; 24/73). The other occupations of the person acquiring the ultrasound images were researchers, radiologists, technicians, paediatricians, biomedical scientists, and medical students. Six studies had individuals who collected ultrasound data who differed from those individuals who interpreted the images/videos for a diagnosis.

Little information was provided on internal validation or external reproducibility. In terms of quality assurance, 9.4% (18/192) of studies detailed their techniques. Eight of these studies validated their data by re-scanning participants, eight by re-reading ultrasound images, one by re-reading ultrasound videos, and one by consultation with a senior radiologist. These are not mutually exclusive: one study both re-scanned participants and re-read images. Further to

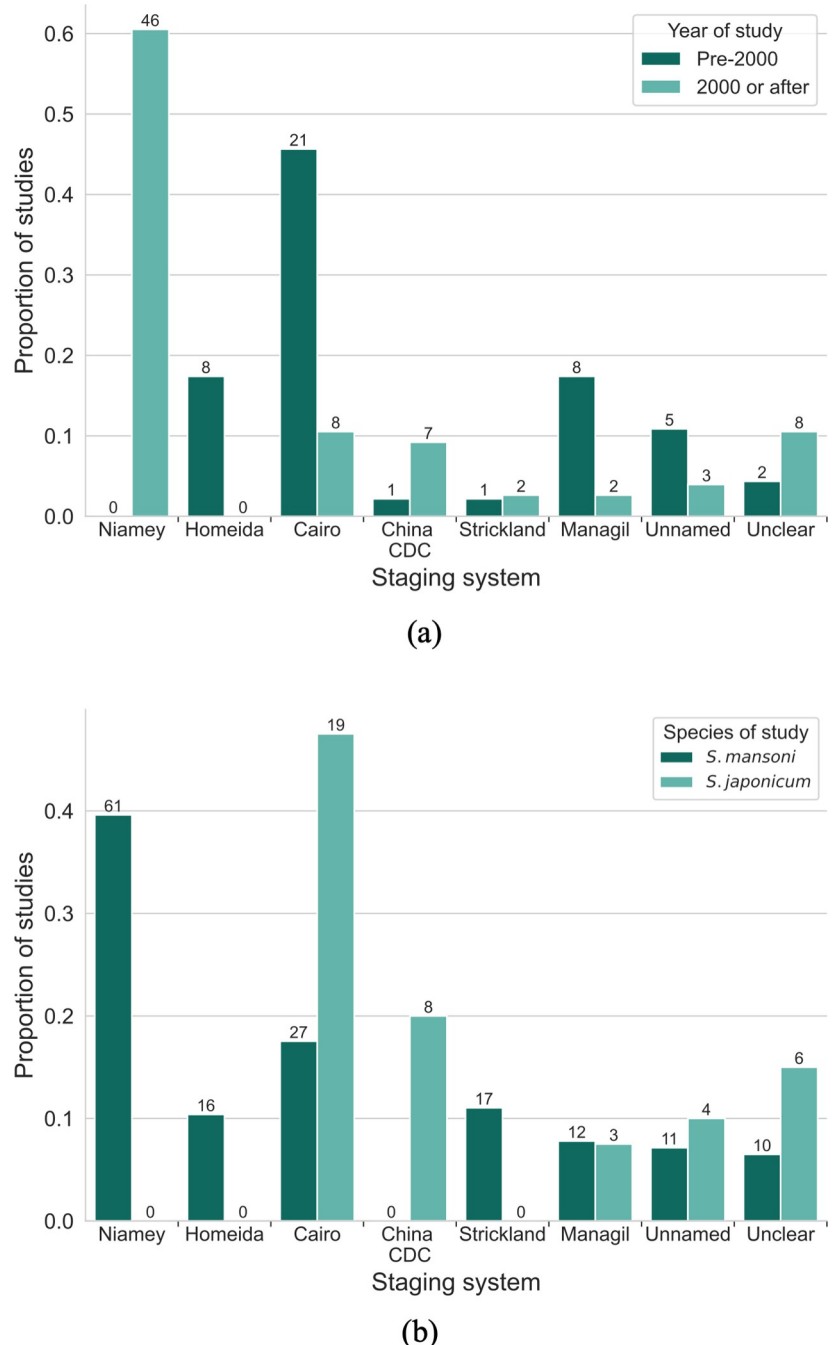

**Fig 4. Use of different staging systems.** A comparison is shown between the use of the five named staging systems and unnamed or unclear staging systems, between (a) study year, and (b) *S. mansoni* and *S. japonicum*. Only studies that stated study year or concerned the relevant species were included in (a) and (b) respectively. The numbers shown above each bar are the numbers of studies that used each protocol. In (a), there is a study that reported a study year before 2000, and also cited the China CDC guidance which was released after 2000. This study is a cross-sectional study design with multiple time-points, and the study year recorded is the year of the first time-point. The China CDC guidance is cited since it is used at a later time-point, after the release of the guideline.

internal validation, 13.0% (25/192) studies mentioned inter- or intra-observer variation of their method. Of these, nine studies made a comment on how inter- or intra-observer variation could impact their results or justify their methods, nine studies cited a different study that

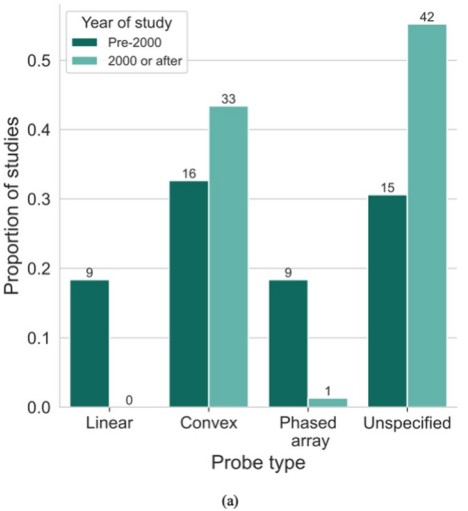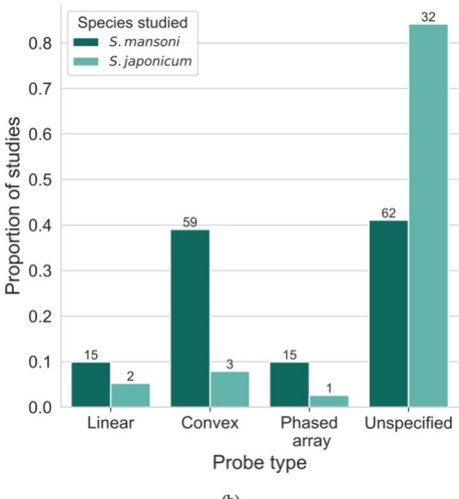

**Fig 5. Type of ultrasound probe used.** A comparison is shown between the use of three different ultrasound probe types and an unspecified probe in the studies, between (a) study year, and (b) *S. mansoni* and *S. japonicum*. The numbers shown above each bar are the numbers of studies that used each ultrasound probe type.

measured the inter- or intra-observer variation of their method, and seven studies measured the inter- or intra-observer variation. Where interobserver variation was quantified for the Managil protocol, the kappa (κ) statistic varied from 0.29–0.5 or slight to moderate agreement. For inter- and intra-observer variation in the image patterns aspect of the Niamey protocol, κ ranged from 0.43–0.89 or moderate to almost perfect agreement. External reproducibility was infrequently mentioned when not confused with internal validation. Only one study compared the same staging method on different populations.

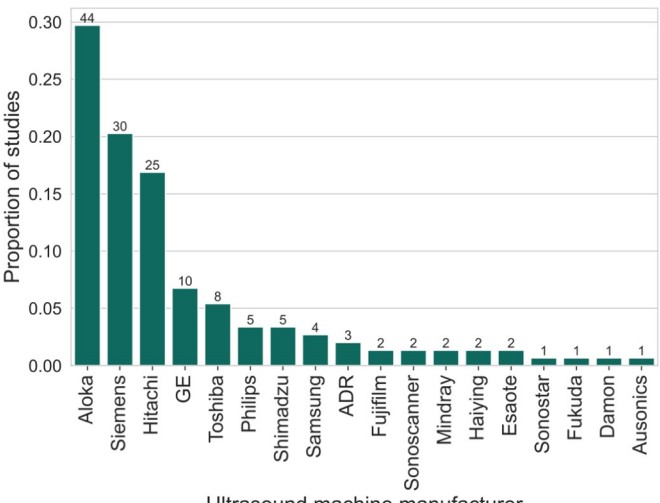

**Fig 6. Manufacturer of ultrasound machine used.** The manufacturers of ultrasound machines used in the studies that reported ultrasound technology are shown. In total, 77.6% (149/192) of studies specified the manufacturer of ultrasound machine used. The machines were not mutually exclusive since some studies used different machines in different sittings or locations of their study. The numbers shown above each bar are the numbers of studies that used each ultrasound machine.

## Discussion

### Reference standards for schistosomal PPF staging

The WHO has proposed that schistosomiasis should be eliminated as a public health problem by 2030 [19]. Central to this target is the control of severe morbidity, which for hepatic schistosomiasis requires accurate staging of schistosomal PPF. This review investigates the use of ultrasound staging systems for schistosomal PPF by synthesising 192 studies using different staging systems across time. Our review found no evidence for an existing gold standard for schistosomal PPF staging and presents support for continuing with ultrasound as a reference standard, while elucidating considerations for protocol development and validation.

This analysis highlights the difficulty in choosing a reference standard—an objective correct representation—for schistosomal PPF and its staging. Ultrasound is often referred to as the gold standard; however, this term has been criticised in the medical sciences due to its implication that this standard represents absolute truth [20]. Although a gold standard may not be perfect, it must be reliable and thoroughly tested, and shown to be internally and externally reproducible in the diagnostic output [21]. Liver biopsy was considered the reference standard for grading before ultrasound imaging was suggested [22]. When ultrasound imaging was proposed as a more sensitive method due to its ability to capture diffuse pathologies of the liver [23], it took between 15–20 years for it to replace biopsy as the agreed-upon reference standard. In addition to increased sensitivity, there are ethical reasons to support replacing biopsy by ultrasound imaging. For liver biopsy, the need for a surgeon and blood transfusion to be easily accessible in the case of complication is often not met in resource-constrained settings [24]. Where reported in studies looking at diagnostic agreement, agreement between liver biopsy and ultrasound imaging was generally good. Serum biomarkers have been considered as a potential alternative reference standard to clinically validate ultrasound staging [25]. However, specific biomarkers for schistosomal PPF have yet to be identified, and this lack of consensus is reflected in the studies in the review which consider a range of biomarkers [26–28]. Also, liver disease is often compensated, that is, liver function is not affected until very late stages of disease. Hence, biomarkers tied to liver function cannot be used as a reference standard in the early stages of disease. There is significant motivation to adopt non-invasive methods, and indeed there now seems to be a consensus that ultrasound is the best available reference standard for schistosomal PPF staging [29,30].

Other non-invasive imaging technologies have been compared to ultrasound imaging for schistosomal PPF staging. Four studies compared MRI to ultrasound imaging [31–34]. However, these studies were conducted nearly 20 years ago (2004–2006) and were on a small scale, ranging from 14–60 participants. More recent studies are possibly absent due to the practical difficulties and high cost of conducting MRI studies in the typical rural healthcare setting where hepatic schistosomiasis is endemic, whereas ultrasound is portable and lower cost. Further, it has been reported that MRI does not correlate consistently well with ultrasound findings when attempting to use the Niamey protocol image patterns for grading [31,34]. Other ultrasound-based techniques that quantify and visualise liver stiffness such as shear-wave elastography [35] and transient elastography [36] have been compared with abdominal ultrasound imaging for schistosomal PPF staging. However, elastography has not been adopted widely, and was only reported in seven studies in this review, perhaps due to its relatively recent availability and the difficulty of acquisition. Additionally, agreement between transient elastography and ultrasound was poor. Therefore, despite existence of alternative non-invasive imaging techniques for schistosomal PPF staging, abdominal ultrasound continues to be the dominant diagnostic reference standard. Assessing the performance of ultrasound against the choice of another technology was not possible in this review due to the heterogeneity in the protocols

used, the populations being studied, and the unvalidated nature of the comparative technology.

## Considerations for ultrasound protocol development

The two current most commonly used protocols to-date evolved from the Cairo protocol. Before 2000, measurement-based and feature-based ultrasound staging systems were used. The Cairo protocol was the most used staging system before 2000; this popularity may be because it was the first staging system recommended by the WHO and there was an increase in the number of studies conducted in the years immediately after publication of the protocol. The Cairo protocol has distinct guidance for *S. mansoni* and *S. japonicum*, with the *S. mansoni* staging system based on three measurements of portal branches and the *S. japonicum* system being feature-based. In 2000, two new sets of guidance were introduced: the Niamey protocol for the staging of *S. mansoni* (in addition to guidance for urinary schistosomiasis) and guidance from the China CDC for the staging of *S. japonicum*. There are common characteristics that are considered in the Cairo and Niamey protocols such as periportal thickening, size of left and right liver lobes, portal vein diameter, presence of collateral vessels and gallbladder wall thickening. The key differences are a mention in the Niamey protocol that measurements of organ size should be height-adjusted and there should be reference measurements from the same population sub-group as well as the addition of PPF image patterns. For the China CDC protocol, the descriptions of each stage of PPF are modified directly from the Cairo protocol, in fact they are identical with some additions. For example, for the third stage of fibrosis, the China CDC guidance adds that the size of the liver is shrunken, the portal vein wall is thickened prominently, and that the blood vessel lumen in the liver is thin and narrow. The China CDC protocol also explicitly recommends the acquisition of the views of the liver from the Cairo protocol.

After 2000, studies conducted in *S. mansoni* endemic countries overwhelmingly adopted the Niamey protocol. Studies on *S. japonicum* either continued to use the Cairo guidance or moved to the China CDC guidance; which was used in eight studies [36–43]. No studies in *S. japonicum* endemic countries adopted the Niamey protocol; this may be due to the lack of staging for *S. japonicum* specific morbidity within the guidance. The Niamey protocol itself recommends the Cairo protocol to stage *S. japonicum* morbidity and suggests a review on standardisation for this species. The high adoption of the Niamey protocol for *S. mansoni* studies suggests that it is still relevant and informing data collection 20 years after from its release.

Looking specifically at *S. japonicum*, it has been suggested that a similar guideline to the Niamey protocol is needed for this species [9]. Yet, guidance has been available from the China CDC since the year 2000 [8]. Due to this being government guidance in a country that does not have English as an official language, the China CDC guidance has not been publicly available in English. We have provided a translation of the descriptions of each grade; however, a full translation of the China CDC handbook to English would be beneficial to non-Mandarin speakers. It is recommended that this book is translated for future use and validation in other *S. japonicum* endemic settings such as the Philippines where it remains a major public health problem.

There are two main components to the Niamey protocol: measurements such as of the portal vein diameter, and image patterns that provide a qualitative representation of different stages of periportal fibrosis. The intent in the guidance is to combine these into a score that grades fibrosis. However, the usage of the complete protocol in the literature was low. Over 75% (49/63) of studies that used the Niamey protocol only used the image patterns element and did not incorporate measurements into the assessment of PPF. Over time, measurement-

based systems have become less often used. A two-site study conducted in Egypt and Kenya [44] reported that measurements of portal vein diameter and portal branch wall thickness were a poor proxy indicator for severe morbidity among different populations, high inter-observer variation existed even when a standardised scanning protocol was used, and height adjustments proved ineffective in improving correlation between scores. After this evidence was published in 2003, studies appeared to use the image patterns proposed in the Niamey protocol near-exclusively as a more internally valid and externally reproducible, as well as easier to intuitively explain and to acquire method for morbidity prediction.

Both the staging systems for *S. mansoni* and *S. japonicum* require views of the abdomen to be acquired for schistosomal PPF staging. The most appropriate probe for abdominal ultrasound of convex, linear and phased-array probes is the convex probe, due to its large coverage with a relatively small footprint (surface area in contact with the skin) [45]. In studies conducted before 2000, a lower proportion of studies used a convex probe than after 2000. The probe type may have affected the way diagnoses were made and how staging systems were applied. Additionally, the frequency of the probe used could affect how diagnoses were made. An increase in frequency of the probe increases the spatial resolution of the image but decreases the penetration of the ultrasound waves. Different frequencies might be used for different purposes such as higher frequencies being used for children since they have a thin abdominal wall and thinner subcutaneous tissue than adults [46,47]. Differential use of frequencies for adults and children has been reported in studies in the review [48], however the reporting of different frequencies was not always clear. Sometimes, a frequency range was reported, for example 2-5MHz. Although a single frequency (for example 3.5MHz) is chosen by the operator when acquiring ultrasound images, ultrasound waves with a range of frequencies typically following a Gaussian-like distribution around a single value are emitted. The different ways to report frequency caused difficulty in comparing technology usage. In terms of recommendations for ultrasound technology, it could be made explicit in the guidance that a convex probe is the most appropriate for implementation of the image patterns section of the Niamey protocol. Also, more consistent reporting of the frequency specification of the probe used is needed to ensure that the abdominal images are being collected at the optimal resolution. None of the studies used more than one machine to allow a within study comparison, and so an evidence-based recommendation for a specific machine cannot be made. The most important distinction between ultrasound machines is whether they are static or portable. Portable machines are widely used in low-resource settings for logistical and cost reasons; however, this comes at a trade-off with image quality [49]. Studies should also report the model specification of the ultrasound machine used. Future studies could explore whether guidance should be provided for using different frequencies for different age groups and compare the use of portable and static ultrasound machines for the staging of schistosomal PPF.

The image patterns aspect of the Niamey protocol has been shown to have κ statistics that are in the moderate to almost perfect intra- or inter-observer agreement ranges of values [44,50,51], while the Managil protocol has been shown to produce fair to moderate values for intra- or inter-observer agreements [52]. There were no κ statistics available for inter- or intra-observer variation in *S. japonicum* studies. It may be concluded that the image patterns aspect of the Niamey protocol is at least moderately internally valid, and in fact such validation has been used as a justification for its use [53]. However, the wide range in κ values gained from the use of the image patterns in a small number of studies shows that this evidence is not conclusive and further studies are needed. One reason for the variation may be because guidance was not always followed as written; modifications were made to the Niamey protocol to fit the purpose of a study. Another source of variation may be that the Niamey protocol lacks guidance on how different abdominal ultrasound views associate to the acquisition of the image

patterns and therefore the individual stages of schistosomal PPF. The protocol also requires further instructions on how to standardise probe sweeps of the abdomen. Studies are currently underway in the SchistoTrack Cohort [54], a prospective cohort study on chronic schistosome infection and schistosomal PPF in Uganda, to create more specific and standardised guidance on how to apply the Niamey protocol.

Future guidance will likely need to consider the application of machine learning to schistosomal PPF staging. Although currently there is no literature in this specific field, machine learning has been applied to other areas of schistosomiasis research and to other forms of liver fibrosis, with more broad artificial intelligence (AI) guidance recently released by the WHO [55]. In schistosomiasis, machine learning has been used for egg counting and morphology [56], automated reading for POC-CCA diagnostic tests [57] and interpretation of images of female genital schistosomiasis [58]. For ultrasound and liver disease, the focus has been on non-alcoholic fatty liver disease (NAFLD) and liver cancer [59,60]. There is ongoing work to develop a tool for automated staging of schistosomal PPF in the SchistoTrack Cohort [61]. Clear standardisation in protocols for schistosomal PPF staging would generate consistent datasets that are suitable for training of machine learning models. Protocols should be designed to reliably support clinicians to identify relationships between information in ultrasound images and clinical outcomes, so that machine learning models can be relied upon to do the same.

## Clinical translation

Establishing who to target for implementation and identifying the use cases for ultrasound staging systems are important considerations when putting together guidance to stage schistosomal PPF. Twenty-nine of the studies in this review considered some link between ultrasound findings of schistosomal PPF and clinical severity, for example prediction of oesophageal varices [62–64] or portal hypertension [50,65]. These concrete clinical outcomes can be linked to ultrasound staging systems and clinically justify their use, establishing prognostic associations for staging. More research is needed to understand how these different schistosomal PPF grade definitions translate into measures of clinical severity. By contrast, other studies used the grade of periportal fibrosis as determined by ultrasound as their outcome. These studies assessed risk factors that predict schistosomal PPF grade [51,66] or measured the effect of some intervention [52,67]. These two distinct groups show that among the research community, there are different uses of ultrasound to stage schistosomal PPF including staging for treatment, surveillance for PPF risk, and early-stage prediction of severe morbidity. These use cases may require different updates to the staging guidance. If these protocols are to be used in clinical practice, rather than in research settings, then a still different approach may be needed.

The distinct use cases of ultrasound staging of schistosomal PPF further highlight the importance of improving standardisation in the development, application, and validation of protocol usage. Twenty-eight studies used ultrasound imaging to measure the effects of some intervention such as praziquantel treatment on fibrosis. In this context, if measurement bias exists that is inconsistent pre- and post-intervention then the classification of schistosomal PPF grade will be incomparable, and the impact of the intervention cannot be evaluated. More research is needed to make a full assessment of the use cases of ultrasound in schistosomal PPF grading in the context of monitoring, evaluation, surveillance and elimination of hepatic schistosomiasis as a public health issue. An inspiration for this research could be the 'Target Product Profile' published by the WHO for schistosome infection diagnostics [53,68], along with WHO guidance on diagnostic ultrasound [54,69]. Future studies could investigate developing a target product profile for ultrasound staging of schistosomal PPF, establishing a reference

standard to improve control of hepatic schistosomiasis-associated morbidity. This research may provide more consistent measures of morbidity, which could better inform global standards used for schistosomiasis disability weights and improve global burden of disease estimates.

## Limitations

There are some limitations of this scoping review. Articles written in languages other than English or Mandarin were excluded. There was a lack of studies that concerned *S. mekongi*, however there were no excluded studies that concerned this species. Therefore, it can be concluded that there is extremely limited literature on *S. mekongi*. This may be due to a lower prevalence or less research being done in the affected countries (Cambodia, Laos and Thailand). As is true for all scoping reviews, there was a reliance on complete and accurate reporting of studies. Most of the variables that were extracted relied on detailed reporting of the methods used, which was perhaps an unfair expectation when ultrasound was not a large part of a study. Therefore, there will be trends in practice that cannot be summarised. The literature also had gaps on key information expected to be reported in a study such as the year of data collection. In order to be able to carry out comparisons between studies, key study characteristics such as study year, the staging system used and any modifications made to this staging system, views of the abdomen collected, quality control techniques and the ultrasound technology used should all be clearly and accurately stated. The nature of a scoping review means that there were no mandatory quality checks on the studies being included in the analysis. If a meta-analysis were to be explored in future work, a risk of bias assessment would need to be conducted to evaluate the studies included here.

## Conclusion

In this review, a framework has been presented for the development of ultrasound staging systems for schistosomal PPF. As well as defining a framework for what an ultrasound staging system should provide, this review has identified research priorities for protocol development and study reporting recommendations. Although the Niamey protocol patterns were most widely used for the staging of schistosomal PPF, this commonality does not necessarily equate to a consensus that the Niamey protocol constitutes a well-defined and reliable reference protocol. For a reference protocol, there are three key aspects: 1) a clear definition of what must be observed to qualify a patient for a disease diagnosis and its severity (a staging system); 2) guidance on where and how these observations should be obtained (well-defined views that contain certain anatomy related to the stages); and 3) a recommendation on what equipment should be used to acquire these observations and what minimal level of expertise is required (e.g., a certain probe or type of ultrasound machine and level of technical qualification). The goal of a reference protocol should be to reproduce diagnoses in different settings. Research still is needed on the external reproducibility of the Niamey protocol image patterns, choice of frequency for ultrasound probes used in children and adults, influence of portable and static ultrasound machines on the interpretation of the image patterns, and variation in ultrasound diagnoses derived from individuals with varying levels of sonography expertise. The last element is particularly important in resource-constrained contexts where high levels of sonography expertise that are currently required to execute existing ultrasound protocols may be unavailable.

Ultrasound imaging has been and remains an important method for schistosomal PPF diagnosis and staging in research. The most commonly used guidance for *S. mansoni*, the Niamey protocol, is no longer being interpreted in the way it was originally intended, which leaves

ambiguity to the end user, and ignores advances in ultrasound technology since the original guidance was written. For these reasons, an update to the Niamey protocol is needed. Ultrasound protocol updates for hepatic schistosomiasis are required to enable ultrasound imaging to be used within surveillance strategies to support meeting goals of elimination of schistosomiasis as a public health problem.

## Supporting information

**S1 Dataset. Extraction table.** This contains the extracted variables referenced in the review for all 192 studies.
(XLSX)

**S1 Table. Table of database hits.** This contains the breakdown of hits for each search term and the full search string in each database that was searched.
(DOCX)

**S1 Text. PRISMA-ScR checklist.** This contains the page numbers in this document of each element of the PRISMA-ScR checklist for scoping reviews.
(DOCX)

**S2 Text. Reference list of extracted studies.** References for all studies extracted as part of the review.
(DOCX)

**S3 Text. China CDC *S. japonicum* ultrasound guidance.** The English translation of ultrasound protocol produced by the China CDC is provided. This is from pages 97–98 of the Chinese Schistosomiasis Control Manual.
(DOCX)

## Acknowledgments

We thank Nia Roberts and George White at the Bodleian Libraries for their help during the initial stages of the review. We thank Imogen Ockenden for her help locating full text articles. We appreciate support and general advice from Fabian Reitzug. SchistoTrack sonographers Simon Mpooya, Victor Anguajibi, Timothy Mugume and Benjamin Ntegeka provided useful insight for understanding the Niamey protocol. We also thank the Chami Group and Noble Group for their feedback after group meetings and presentations.

## Author Contributions

**Conceptualization:** Eloise S. Ockenden, Goylette F. Chami.

**Data curation:** Eloise S. Ockenden, Sandrena Ruth Frischer, Huike Cheng.

**Formal analysis:** Eloise S. Ockenden.

**Funding acquisition:** Eloise S. Ockenden, Goylette F. Chami.

**Investigation:** Eloise S. Ockenden.

**Methodology:** Eloise S. Ockenden.

**Resources:** J. Alison Noble, Goylette F. Chami.

**Supervision:** J. Alison Noble, Goylette F. Chami.

**Validation:** Eloise S. Ockenden, Sandrena Ruth Frischer, Huike Cheng.

**Visualization:** Eloise S. Ockenden.

**Writing – original draft:** Eloise S. Ockenden.

**Writing – review & editing:** Eloise S. Ockenden, Sandrena Ruth Frischer, Huike Cheng, J. Alison Noble, Goylette F. Chami.

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
