## [Decision Letter · Decision Letter 0]

20 Nov 2023

Dear Dr Chami,

Thank you very much for submitting your manuscript "The role of point-of-care ultrasound in the assessment of schistosomiasis-induced liver fibrosis: a systematic scoping review" for consideration at PLOS Neglected Tropical Diseases. As with all papers reviewed by the journal, your manuscript was reviewed by members of the editorial board and by several independent reviewers. In light of the reviews (below this email), we would like to invite the resubmission of a significantly-revised version that takes into account the reviewers' comments. 

The reviewers agreed that this is an important, well-justified scoping review. Given the large number of studies compiled, there is a request for more synthesis of the information compiled in the results (perhaps shunting some to supplementary material).

We cannot make any decision about publication until we have seen the revised manuscript and your response to the reviewers' comments. Your revised manuscript is also likely to be sent to reviewers for further evaluation.

Sincerely,

Elizabeth J Carlton

Academic Editor

Eva Clark

Section Editor

The reviewers agreed that this is an important, well-justified scoping review. Given the large number of studies compiled, there is a request for more synthesis of the information compiled in the results (perhaps shunting some to supplementary material).

Reviewer's Responses to Questions

**Key Review Criteria Required for Acceptance?**

**Methods**

-Are the objectives of the study clearly articulated with a clear testable hypothesis stated?

-Is the study design appropriate to address the stated objectives?

-Is the population clearly described and appropriate for the hypothesis being tested?

-Is the sample size sufficient to ensure adequate power to address the hypothesis being tested?

-Were correct statistical analysis used to support conclusions?

-Are there concerns about ethical or regulatory requirements being met?

Reviewer #1: 1. Authors should include a Chinese or Chinese-speaking author to include the 29 excluded manuscripts in Chinese, which will more than double the articles pertaining to ultrasound (US) use in S. japonicum, and to appropriately describe the China CDC US guidelines 

2. Describe the excluded 34 manuscripts without full text and how it affects the review

Reviewer #2: Study is well described and the value of performing a scoping review is well articulated. The study design appropriately addresses the stated objective and question stated. The article selection criteria, review and selection are well described. No ethical considerations in this review.

**Results**

-Does the analysis presented match the analysis plan?

-Are the results clearly and completely presented?

-Are the figures (Tables, Images) of sufficient quality for clarity?

Reviewer #1: 1. Overall, there is a limited analytical description of the studies, focusing only on their general features. Some sections of the Discussion can be expounded in the Results (Clinical translation, intra/inter-observer agreement). A brief mention in the Results on the performance of ultrasound as a diagnostic test for PPF, a synthesis on how the Cairo protocol evolved into the Niamey protocol, and elaboration on why Niamey is not being used for S. japonicum are suggested

2. Table 1 shows that most studies focused on diagnostic agreement, but there were minimal details. Were staging systems being compared, or other procedures (Lines 296-9)? Elaboration on the performance of US on clinical severity and risk factors is also suggested 

3. Please clarify Fig 4 and Line 305 for consistency on number of studies utilizing the Niamey protocol

4. Was data available to assess whether certain ultrasound machines perform better than others?

Reviewer #2: Results are clearly presented. Given the quantity of data that the authors abstracted there is only a limited amount of this data presented. It would be valuable to add some additional data analysis. I would suggest looking at who is obtaining the ultrasound images. I realize not all of the studies report on the sonographer but this would be valuable information to add for the studies that do report this.

**Conclusions**

-Are the conclusions supported by the data presented?

-Are the limitations of analysis clearly described?

-Do the authors discuss how these data can be helpful to advance our understanding of the topic under study?

-Is public health relevance addressed?

Reviewer #1: The conclusions are sound, however the statement that there is "difficulty in choosing a reference standard for schistosomal PPF staging" (378) may be unsupported given that 70% of the studies on S. mansoni after 2000 have adopted the WHO-recommended Niamey protocol.

Reviewer #2: Conclusion is well supported by the data presented. The authors are clear in discussing what the data suggests about the current state of ultrasound for intestinal schisto and link this to next steps in how to advance our understanding in this area. Public health value is clearly addressed.

**Editorial and Data Presentation Modifications?**

Reviewer #1: The manuscript is easy to follow and understand

Reviewer #2: Minor revisions as outlined in comments.

**Summary and General Comments**

Reviewer #1: The manuscript is an expansive review of the literature on ultrasound for hepatosplenic schistosomiasis. The manuscripts describes the different aims, populations, study settings and species of studies on ultrasound for schistosomiasis. A historical perspective of the various staging platforms is noteworthy.

Reviewer #2: In this study authors perform a scoping review of the current literature on the use of ultrasound for intestinal schistosomiasis and heterogeneity in what type of protocol (if any) studies adhere to. This is a well done scoping review that presents important data and highlights a known but significant gap in the application of ultrasound for this disease pathology. As noted above the quantity of data the authors abstracted could allow or additional data around image acquisition specifics to be reported in the results section. 

The authors should expand their limitations section especially around Mekongi and Japonicum species - given the number of mandarin studies that were excluded this scoping review primarily tells us about Mansoni studies. This does not diminish the author’s conclusions but it is worth highlighting especially if new consensus about ultrasound protocols are to be considered. 

The biggest gap in this study that needs to be addressed is the area of machine learning. The authors do not even consider this application and their study lacks future relevance by excluding this topic. There is a growing use of machine learning and “auto grading” of imaging studies by computers. More than the need to standardize for human reviewers there is an urgent need to create a consensus around schistosomiasis scoring systems before computers are programmed to read and interpret these images based on an outdated protocol. The authors should expand their conclusion to address this issue moreover concisely and directly. A brief review to see if any studies have already been published around machine learning or AI and schisto would add strength to the study conclusions and the call for an updated consensus protocol.

PLOS authors have the option to publish the peer review history of their article (what does this mean?). If published, this will include your full peer review and any attached files.

Reviewer #1: No

Reviewer #2: No

Figure Files:

Data Requirements:

Please note that, as a condition of publication, PLOS' data policy requires that you make available all data used to draw the conclusions outlined in your manuscript. Data must be deposited in an appropriate repository, included within the body of the manuscript, or uploaded as supporting information. This includes all numerical values that were used to generate graphs, histograms etc.. For an example see here: http://www.plosbiology.org/article/info:doi%2F10.1371%2Fjournal.pbio.1001908#s5.
---

## [Decision Letter · Decision Letter 1]

28 Feb 2024

Dear Dr Chami,

We are pleased to inform you that your manuscript 'The role of point-of-care ultrasound in the assessment of schistosomiasis-induced liver fibrosis: a systematic scoping review' has been provisionally accepted for publication in PLOS Neglected Tropical Diseases.

Best regards,

Elizabeth J Carlton

Academic Editor

Eva Clark

Section Editor

<style type="text/css">p.p1 {margin: 0.0px 0.0px 0.0px 0.0px; line-height: 16.0px; font: 14.0px Arial; color: #323333; -webkit-text-stroke: #323333}span.s1 {font-kerning: none

</style>

Reviewer's Responses to Questions

**Key Review Criteria Required for Acceptance?**

**Methods**

-Are the objectives of the study clearly articulated with a clear testable hypothesis stated?

-Is the study design appropriate to address the stated objectives?

-Is the population clearly described and appropriate for the hypothesis being tested?

-Is the sample size sufficient to ensure adequate power to address the hypothesis being tested?

-Were correct statistical analysis used to support conclusions?

-Are there concerns about ethical or regulatory requirements being met?

Reviewer #2: Methods are well described including searching multiple databases and what data was extracted. The only portion that is less clear is the specific data extraction tool or checklist that was used. Given the heterogenity of the articles extracted it would be helpful for the readers to know what specific tool was used and how the authors decided what the most important pieces of data would be.

**Results**

-Does the analysis presented match the analysis plan?

-Are the results clearly and completely presented?

-Are the figures (Tables, Images) of sufficient quality for clarity?

Reviewer #2: Well described. The included figures add extensively to the written results presented. Many different aspects of ultrasound evaluation of schistosomiasis are reported on. There would be some benefit in the authors highlighting where they feel there results have the strongest support from the articles reviewed. Given that there is no included grading of the quality of the studies the authors opinion on what results seem to have the most robust support would be helpful to the readers.

**Conclusions**

-Are the conclusions supported by the data presented?

-Are the limitations of analysis clearly described?

-Do the authors discuss how these data can be helpful to advance our understanding of the topic under study?

-Is public health relevance addressed?

Reviewer #2: The authors conclusions are well supported by the data they present. They call for a revised system for ultrasound grading of schistosomiasis. There is an opportunity for the authors to make some more targeted suggestions of what the framework of an appropriate clinical and research grading system might look like. There is an ongoing question of how consensus might be reached but

**Editorial and Data Presentation Modifications?**

Reviewer #2: (No Response)

**Summary and General Comments**

Reviewer #2: This paper does the important and hard work of summarizing a large an heterogenous body of literature on ultrasound and schistosomiasis published over the past 40 years. The authors make a strong argument that there is a need for a revised system that helps clinicians stage disease clinically and that helps researchers focused on tracking prevalence and severity of infection in endemic regions.

PLOS authors have the option to publish the peer review history of their article (what does this mean?). If published, this will include your full peer review and any attached files.

Reviewer #2: No

---

## [Editor Report · Acceptance letter]

6 Mar 2024

Dear Dr Chami,

We are delighted to inform you that your manuscript, "The role of point-of-care ultrasound in the assessment of schistosomiasis-induced liver fibrosis: a systematic scoping review," has been formally accepted for publication in PLOS Neglected Tropical Diseases.

Best regards,

Shaden Kamhawi

co-Editor-in-Chief

Paul Brindley

co-Editor-in-Chief
